# Prevalence and associated factors of post-partum depression in Ethiopia. A systematic review and meta-analysis

Tadele Amare Zeleke[1]*, Wondale Getinet[1], Zemenu Tadesse Tessema[2], Kassahun Gebeyehu[3]

1 Department of Psychiatry College of Medicine and Health Science, University of Gondar, Gondar, Ethiopia,
2 Department of Epidemiology and Biostatics, Institute of Public Health College of Medicine and Health Sciences, University of Gondar, Gondar, Ethiopia, 3 Department of Medical Nursing, School of Nursing, College of Medicine and Health Science, University of Gondar, Gondar, Ethiopia

* tadeleamare14@gmail.com

**Data Availability Statement:** All relevant data are within the manuscript and its Supporting Information files.

## Abstract

### Background

Globally, post-partum depression is a major public health problem and is associated with a harmful effect on the infant, child, and mothers' mental, physical, and social health. Although a few post-partum depression studies have been published, we still lack an accurate estimated pooled prevalence of national PPD and associated factors.

### Objectives

This study aims to show the estimated pooled prevalence of PPD and associated factors in Ethiopia.

### Methods

We conduct the extensive search of articles as indicated in the guideline (PRISMA), reporting systematic review and meta-analysis. Databases like MEDLINE, PubMed, psych INFO, Web of Science, EMBASE, CINAHL, Scopus, and The Cochrane Library. All publications and grey literature were addressed by using MeSH terms and keywords. The pooled estimated effect of post-partum depression and associated factors was analyzed using the random effect model meta-analysis, and 95% CI was also considered.

### Protocol and registration

PROSPERO 2020 CRD42020176769 Available from https://www.crd.york.ac.uk/prospero/display_record.php?ID=CRD42020176769.

### Result

A total of 11 studies with 7,582 participants were included in this meta-analysis. The estimated pooled prevalence of post-partum depression (PPD) was 22.08%, with a 95% CI (17.66%, 26.49). For factors associated with post-partum depression, a random effect size

**Funding:** The authors received no specific funding for this work.

**Competing interests:** The authors have declared that no competing interests exist.

model was used during meta-analysis; unplanned pregnancy [(OR = 2.84; 95% CI (2.04, 3.97)], domestic violence [OR = 3.14; 95% CI (2.59, 3.80)], and poor social support [OR = 3.57;95% CI (2.29,5.54) were positively associated factors with post-partum depression.

## Conclusion and recommendation

The estimated pooled prevalence of post-partum depression was high in Ethiopia. Unplanned pregnancy, poor social support, and domestic violence were factors affecting PPD. Therefore, the Ethiopian policymakers and health personnel better give more emphasis to mothers who had a history of unplanned pregnancy, domestic violence, and poor social support

## Introduction

Depression is a major public health problem that women are around twofold greater than men to experience depression during their lifetime [1–3].

Globally, depressive disorders are common, chronic, and a principal source of disability among women. In the US, approximately 12 million women experience clinical depression each year [1].

In low- and middle-income countries, the estimated prevalence of depression among women ranges from 15 to 28% in Asia and Africa [4,5].

Post-partum depression (PPD) is a mood disorder that involves the brain and affects behavior and physical health after delivery [6]. Worldwide, depression is the leading cause of disability [7], and it affects one in five women after giving birth [8]. Post-partum depression is the most common psychiatric illness, and it is the major public problem that is as twice as common in women as men during the childbearing years [9].

Globally, severe postnatal onset depression rates are three times higher than in other periods of women's lives [9]. Parenting stress and mother-child interaction problems are common in postnatal depressive mothers [10].

In the world, depression after delivery affects women. It increases poorer self-care and maternal morbidity and negatively affects infants, children, and families as well [5,11]. The disorder is often unrecognized and untreated, despite the potentially deleterious effects of PPD on the mother, infant, and children [9,12]. Because of the stigma of depression, the mother may refuse to seek professional help [13]. Maternal PPD has short-term negative effects on young children's emotional, cognitive, behavioral, and interpersonal development [9].

In a worldwide review, the prevalence of PPD ranges from 4.0 to 63.9% [14]. In 56 countries, the pooled prevalence of PPD was 17.7% [15]. In India, the systematic review and meta-analysis of the pooled prevalence of PPD was 22% [16]. In another review, the pooled prevalence was 20% [17]. In Iran, a systematic review and meta-analysis, the pooled prevalence was 25.3% [18]. In a systematic review in Denmark, up to 15% of the mother were affected with post-partum depression [19].

In low and middle-income countries, there are large gaps in the knowledge of the long-term effects of post-partum depression in physical, psychological, and social [10] and PPD is under-recognized and under-treated [19].

In different literature reviews; factors associated with PPD were unplanned pregnancy [20–22], a having history of depression [18,23–25], perceived lack of support from husband [25], domestic violence and lack of support [16,22], poor social support from the partner [26], birth

complications [27], dissatisfaction about family [28], violence from husband [29–31] and poor social support [9,23,32].

Although mothers after delivery are at a critical period for the incidence of depression, little attention is still given in terms of prevention and treatment. Showing the pooled prevalence and factors associated with PPD by systematic review and meta-analysis is very important to health policymakers to pertain attention for these vulnerable women. Therefore, the present study reviews accessible epidemiological publications on post-partum depression and related factors in Ethiopian women to help health workers and policymakers design preventive strategies and further research.

## Our two purposes in this study

a.  What is the estimated pooled prevalence of post-partum depression in Ethiopia?

b.  What are the associated factors for post-partum depression in Ethiopia?

**Intervention(s), exposed(s)**; postnatal mothers who considered depression as screened by depression screening tools

**Comparator(s)/control**; postnatal mothers who have considered no depression by depression screening tools

## Materials and methods

### Search process and study selection

Literature search; Our search strategy and selection of publication for the review were conducted according to the PRISMA guideline [33]. The literature on post-partum depression among Ethiopian women was retrieved through searching the scientific search engines Database like MEDLINE, PubMed, psych INFO, Web of Science, EMBASE, CINAHL Scopus, and The Cochrane Library. All publications and grey literature were addressed by using keywords that were used in PubMed. In PubMed, MeSH terms were used (incidence OR prevalence OR magnitude OR epidemiology) AND (postnatal depression OR depression OR post-partum depression OR depressive disorder OR maternal mental health OR emotional distress OR puerperal disorder OR low mood disorder OR psychological distress) AND (after childbirth OR after delivery) AND (associated factors OR risk factors OR predictors OR determinants) AND Ethiopia, January 2010 to January 2020.

For the other databases, we employed specific subject headings as advised for each database. Furthermore, to identify other related literature, we manually searched the reference lists of eligible articles.

**Protocol and registration**; PROSPERO 2020 CRD42020176769 Available from https://www.crd.york.ac.uk/prospero/display_record.php?ID=CRD42020176769.

### Eligible criteria

Inclusion criteria

- Study design type-cross-sectional

- Article published in the English language

- Studies that reported the prevalence of post-partum depression in the health institution and in the community

- A study done in Ethiopia

- Publication date from 1 January 2010 to 1 January 2020

- All publications which fulfilled more than 90% of the criteria were included

**Exclusion criteria.** Reviews, letters and international studies, and duplicated studies were excluded.

## Methods for data extraction and quality assessment

Three reviewers (TAZ, WG, and ZTT) evaluated the relevant articles using the title and the abstract prior to retrieval of the full-text articles. Retrieved full-text articles were further screened according to prespecified inclusion and exclusion criteria. We resolved the arguments by a discussion with the fourth reviewer (KG).

The standardized form of the data extraction method was used for identified studies. The following information was extracted for each included study: first author, year of publication, study design, associated factors, sample size, study settings, adjusted for risk estimate (OR), and the 95% confidence interval. Data extraction from source documents was done independently by four investigators. The disagreement was resolved with discussion.

The quality of the included studies was evaluated using the Newcastle-Ottawa Scale(NOS) [34]. Sample representativeness and size, comparability between participants, ascertainment of post-partum depression, and statistical quality were the domains of NOS used to assess each study's quality. Actual agreement and agreement beyond chance (unweighted Kappa) were used to evaluate four reviewers' agreement. We consider the value 0 as poor agreement, 0.01 to 0.02 as slight agreement, 0.21 to 0.4 as a fair agreement, 0.41 to 0.60 as moderate agreement, 0.61 to 0.80 as substantial agreement, and 0.81 to 1.00 as almost perfect agreement [35]. In this review, the actual agreement beyond chance was ranged from 0.88 to 1 is almost perfect agreement.

## Data synthesis and analysis

STATA version 14 software was used for meta-analysis. Forest plots that showed combined estimates with 95% CI. The heterogeneity was evaluated using Q and $I^2$ statistics [36]. For the variables, the random effect size (OR) model was used. The magnitude of statistical heterogeneity between studies was assessed using $I^2$ statistics and considered value 25% as low, 50% as a medium, and 75% as high [37]. In this review data, the value of the I2 statistics was 95.1% with a p-value $\leq$ of 0.001, which showed there was high heterogeneity. Therefore, the overall pooled prevalence was estimated by the random effect model meta-analysis [36]. Meta-regression was made to explore the probable source of heterogeneity. We also carried a leave-one-out sensitivity analysis to assess the key studies that significantly impact between-study heterogeneity.

## Variables; post-partum depression yes/no

Independent variables were unplanned pregnancy vs. planned pregnancy, poor social support vs. strong social support, and domestic violence vs. no domestic violence.

## Result

## Identification of the studies

Our search strategy and selection of publication for the review was conducted in accordance with the PRISMA. In the database search, 718 articles were found. Of these, 688 articles were

excluded because the title and the abstract were not fit the inclusion criteria. Eighteen articles were retrieved for full screening. However, seven studies were excluded because there was perinatal depression. Therefore, eleven studies were included in this systematic review and meta-analysis guideline (Fig 1).

## PRISMA 2009 Flow Diagram

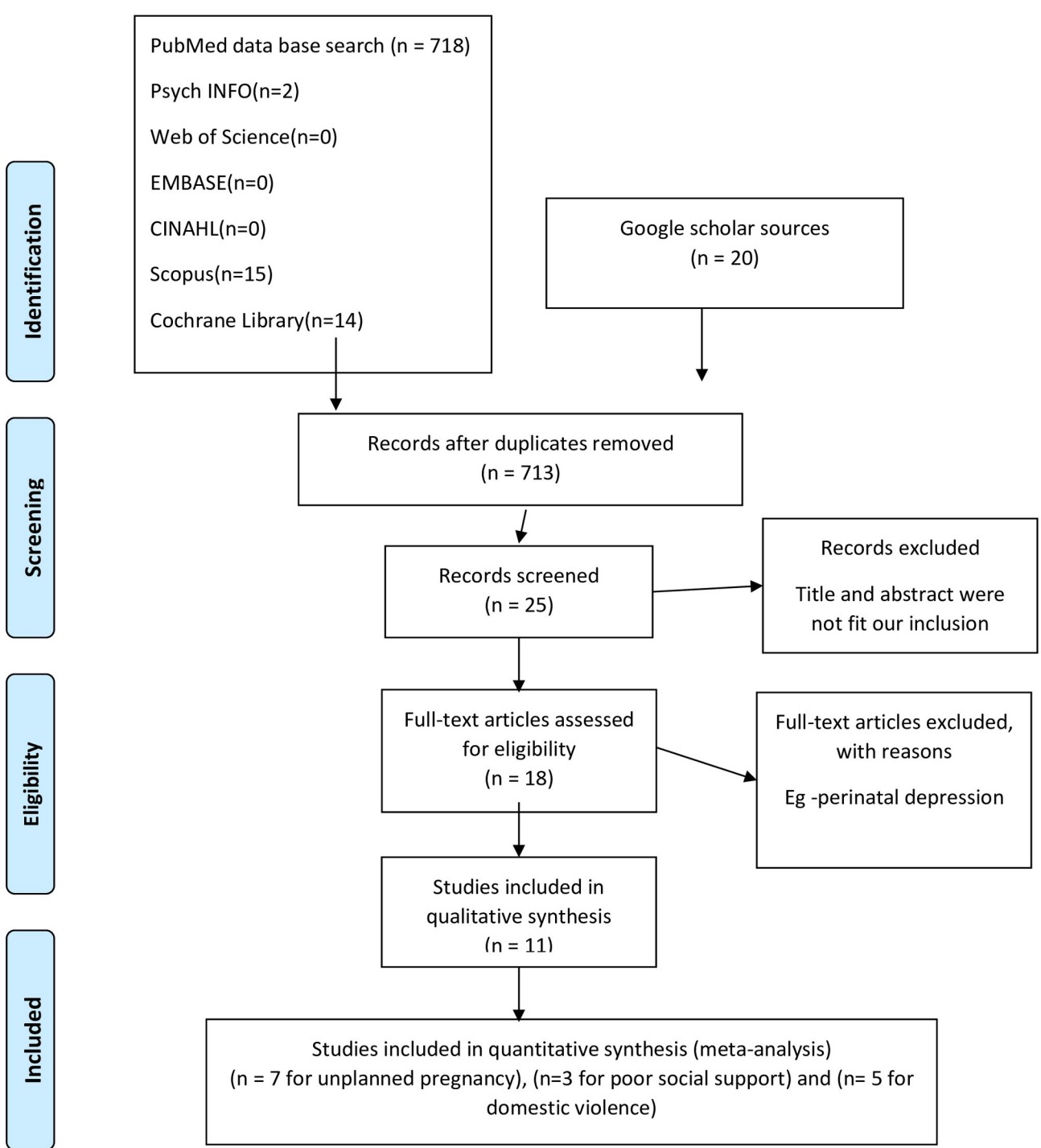

**Fig 1. Flow chart showing how was the research articles were searched, 2020.**

**Table 1. The prevalence of post-partum depression among women related to the study area in Ethiopia, 2020.**

| Author | Publication year | Study area | Site | Study design | PPD | sample | case | prevalence | Tool |
|---|---|---|---|---|---|---|---|---|---|
| Asaye MM. [38] | 2020 | Gondar town | Community | cross-sectional | YES | 526 | 129 | 25 | EPDS |
| Fantahun A. [39] | 2018 | Addis Ababa | Health institution | cross-sectional | YES | 618 | 144 | 23.3 | EPDS |
| Teshome H [40] | 2016 | Addis Ababa | Health institution | cross-sectional | YES | 295 | 82 | 27.8 | K-10 score. |
| Abadiga M. [41] | 2019 | Nekemte Town | Community | cross-sectional | YES | 287 | 60 | 20.9 | EPDS |
| Shewangzaw A. [42] | 2018 | Harar Town | Health institution | cross-sectional | YES | 122 | 16 | 13.11 | EPDS |
| Azale T [43] | 2018 | Sodo district | Community | cross-sectional | YES | 3147 | 385 | 12.2 | PHQ-9 |
| Shitu S. [44] | 2019 | Ankesha district | Community | cross-sectional | YES | 596 | 141 | 23.7 | EPDS |
| Abebe A. [45] | 2019 | Bahir Dar Town | Health institution | cross-sectional | YES | 511 | 113 | 22.1 | EPDS |
| Kerie S. [46] | 2018 | Mizan Tepi Town | Health institution | cross-sectional | YES | 408 | 138 | 33.8 | EPDS |
| Toru T. [47] | 2018 | Mizan Aman Town | Community | cross-sectional | YES | 456 | 102 | 22.4 | PHQ-9 |
| Mariam D. [48] | 2016 | Eastern Zone of Tigray | Community | cross-sectional | YES | 616 | 117 | 19 | EPDS |

## Characteristics of the studies

In this systematic review and meta-analysis, 11 articles were included. The included studies were conducted from 1 January 2010 to 1 January 2020. All studies were conducted with a cross-sectional study design in Ethiopia. Post-partum depression was assessed by using EPDS (eight studies), PHQ-9 (two studies), and K-10 score (one study). Six studies were conducted in the community; the rest were in the health institution (Table 1).

## Qualities of included studies

The Newcastle-Ottawa Scale (NOS) was used to assess the quality of the studies methodologically. In the evaluation, we concluded that 11 studies satisfy the quality assessment in terms of selection, outcome measurement, and non-response bias. The risk of bias in each study was assessed by using kappa values, which range from 0.88 to 1, almost perfect (Table 2).

## Publication bias

No evidence of publication bias was found by the funnel plot and Egger's regression test of post-partum depression (Fig 2).

## Sensitivity analysis

In the sensitivity analysis, there is no single study that is influencing the overall meta-analysis estimate (Fig 3)

**Table 2. The quality and agreed level of bias and level of agreement on the method qualities included articles in a meta-analysis based on sample, outcome, objective, responses rate and analysis method.**

| Author | Publication year | Study area | Percentage of agreement | Kappa value | Level of agreement | NOS quality(0 9) |
|---|---|---|---|---|---|---|
| Asaye MM. | 2020 | Gondar town | 100 | 1 | Almost perfect | 9 |
| Fantahun A. | 2018 | Addis Ababa | 100 | 1 | Almost perfect | 9 |
| Teshome H | 2016 | Addis Ababa | 88 | 0.88 | Almost perfect | 8 |
| Abadiga M. | 2019 | Nekemte Town | 100 | 1 | Almost perfect | 9 |
| Shewangzaw A. | 2018 | Hrarar Town | 100 | 1 | Almost perfect | 9 |
| Azale T | 2018 | Sodo district | 100 | 1 | Almost perfect | 9 |
| Shitu S. | 2019 | Ankesha district | 100 | 1 | Almost perfect | 9 |
| Abebe A. | 2019 | Bahir Dar Town | 100 | 1 | Almost perfect | 9 |
| Kerie S. | 2018 | Mizan Tepi Town | 100 | 1 | Almost perfect | 9 |
| Toru T. | 2018 | Mizan Aman Town | 100 | 1 | Almost perfect | 9 |
| Mariam D. | 2016 | Eastern Zone of Tigray | 100 | 1 | Almost perfect | 9 |

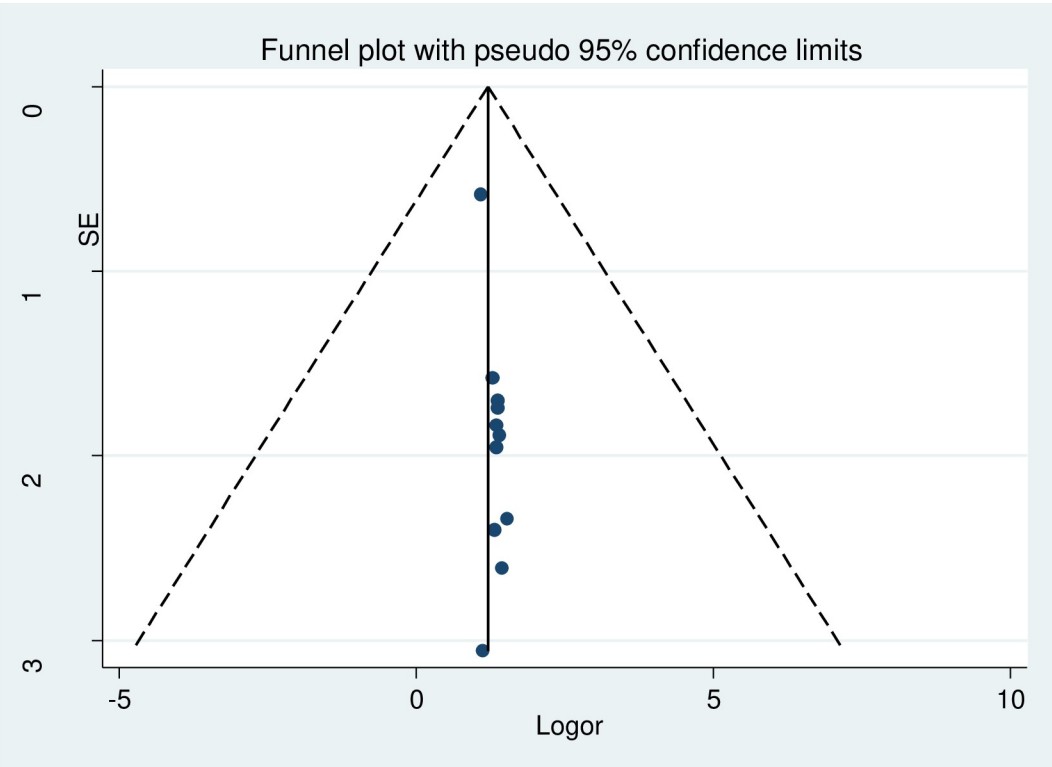

**Fig 2. Forest plot presenting of publication bias of post-partum depression among after child giving mothers, Ethiopia, 2020.**

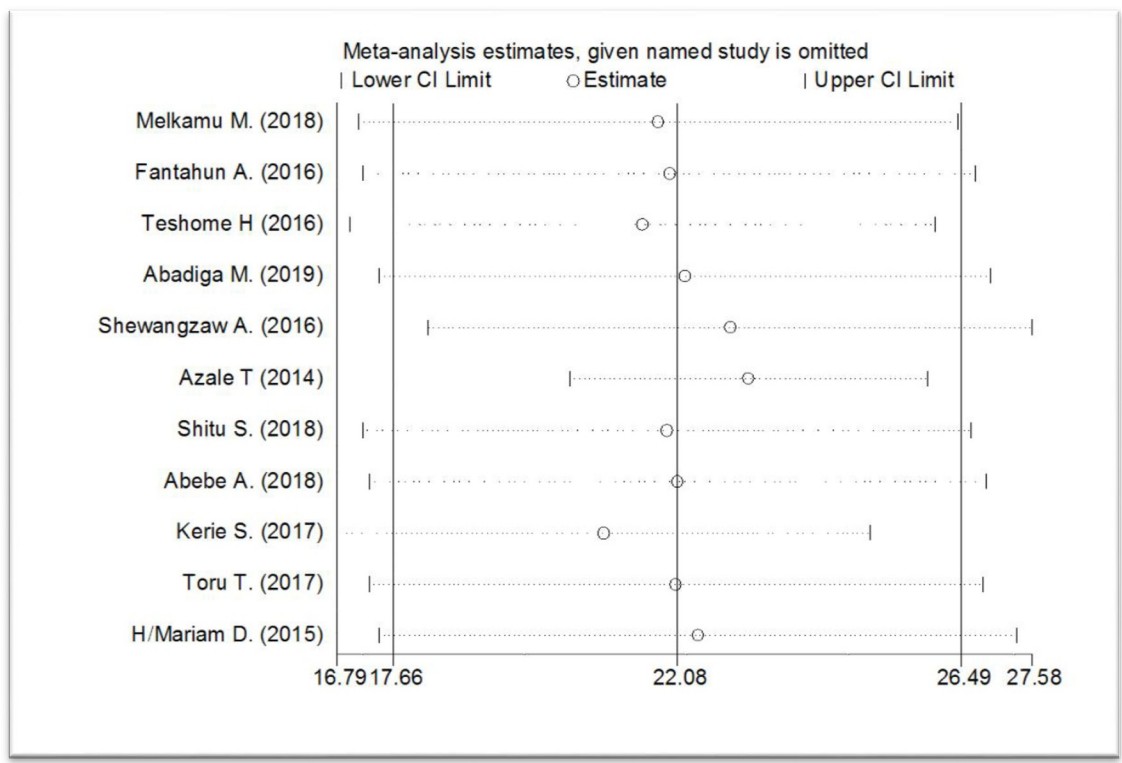

**Fig 3. Forest plot presenting to show not having a single study influences the overall meta-analysis estimated of post-partum depression among after child giving mothers, Ethiopia, 2020.**

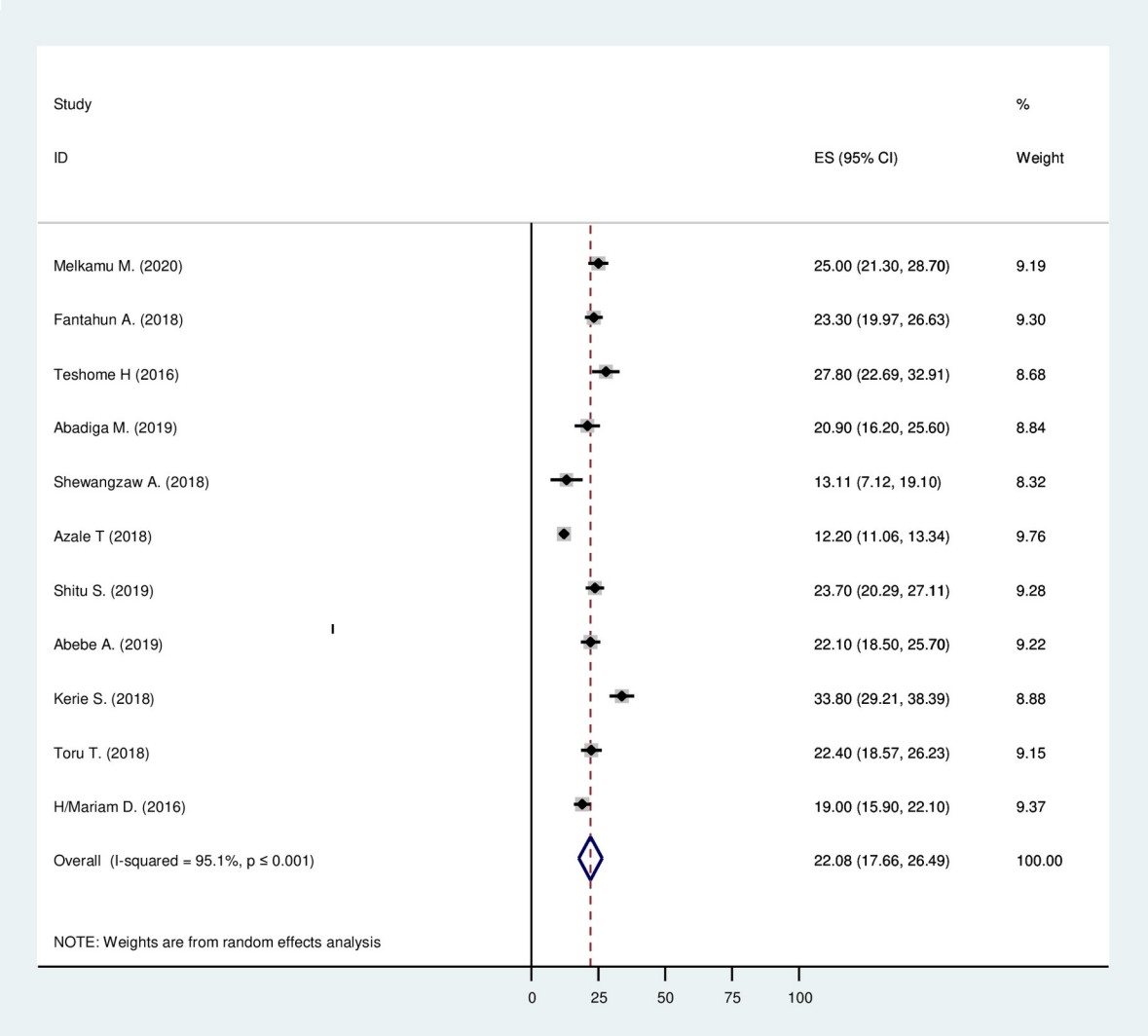

**Fig 4. The estimated pooled prevalence of post-partum depression among mothers after giving birth in Ethiopia, 2020.**

### The result of estimated a pooled meta-analysis

**Prevalence of post-partum depression.** A total of 11 studies with 7,582 participants were included in this meta-analysis. In Ethiopia, the prevalence of post-partum depression was ranged from 12.2% to 33.8% (Table 1). The random-effect model was used to combine the 11 articles to show the estimated pooled prevalence of post-partum depression. The estimated pooled prevalence of post-partum depression (PPD) among mothers was 22.08%, with 95% CI (17.66%, 26.49%). The studies' heterogeneity was significant ($I^2$ = 95.1%; Q = 204.06 df = 10 and p≤0.001) (Fig 4).

**Subgroup analysis by assessment tools.** Around eight articles were conducted with EPDS, two articles were with PHQ-9, and one study was conducted with a K-10 score. The prevalence of PPD by using EPDS, PHQ-9, and K-10 Score was 27.76% with 95% CI (19.46, 26.05), 17.13% with 95% CI (7.14, 27.12), and 27.80% with 95% CI (22.69, 32.91) respectively. The heterogeneity of each tool, EPDS, and PHQ-9, was significant ($I^2$ = 82.7, Q = 40.38, df = 7, p≤0.001), and ($I^2$ = 96.0, Q = 25.06, df = 1, p≤0.001) respectively. In all assessment tools; the

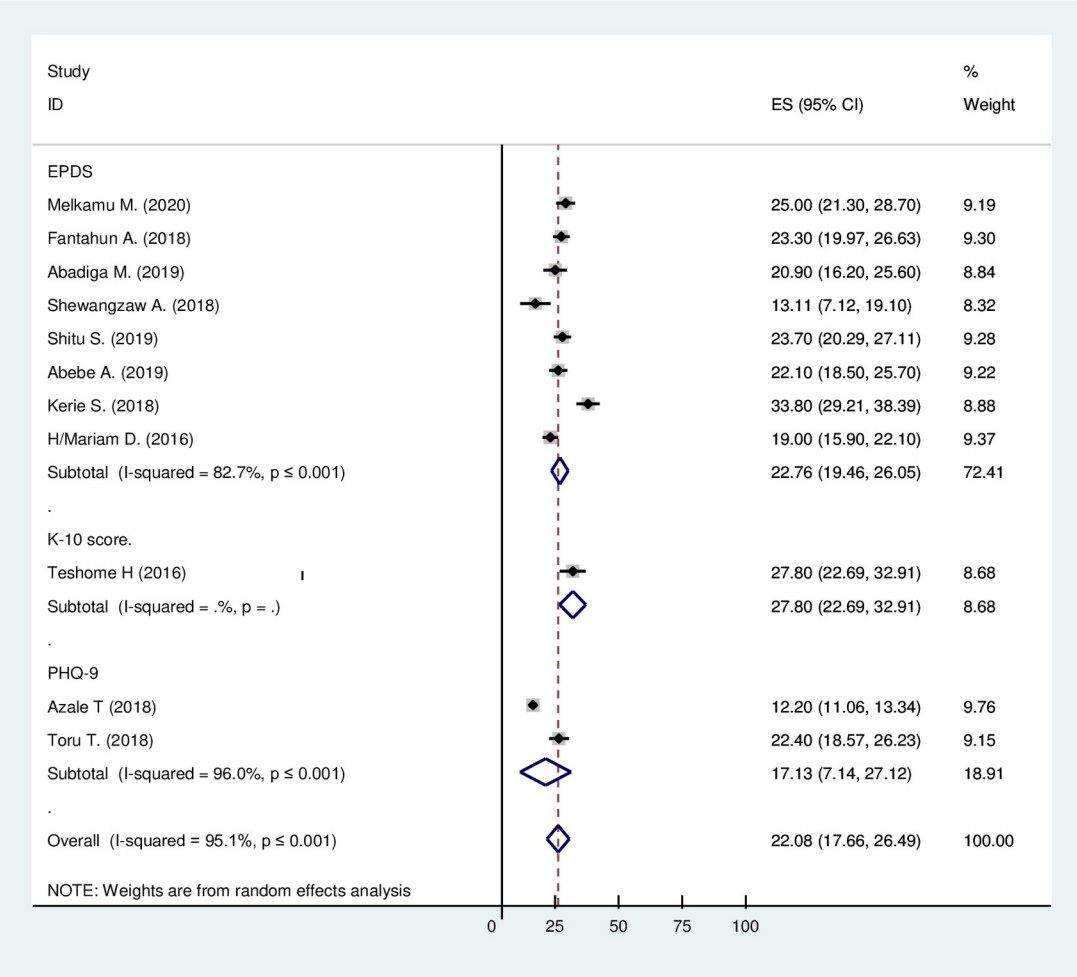

**Fig 5. Forest plot presenting of subgroup analysis of the pooled estimated prevalence of post-partum depression based on tools in Ethiopia, 2020.**

prevalence of PDD were similar. Since the study conducted by the K-10 assessment tool was a single study, there was no heterogeneity test (Fig 5).

**Unplanned pregnancy and post-partum depression.** From (Fig 6) a total of seven articles were included in the analysis. There was a significant association between unplanned pregnancy and post-partum depression. Mothers who had a history of unplanned pregnancy were about 3(OR = 2.84; 85% CI 2.04 to 3.97) times more likely to have depression when compared to a planned pregnancy.

**Social support and post-partum depression.** Three studies were carried out in this meta-analysis. The pooled odds ratio (OR) demonstrated that the odds of PPD were significantly higher in mothers who had poor social support than mothers' who had strong social support (OR = 3.57; 95% CI 2.29 to 5.54) (Fig 7).

**Domestic violence and post-partum depression.** In (Fig 8) a total of five articles were comprised in this analysis. There was a significant association between domestic violence and post-partum depression. Having domestic violence was about 3 (OR = 3.14; 95% CI 2.59, 3.80) times more likely to have post-partum depression than not having domestic violence.

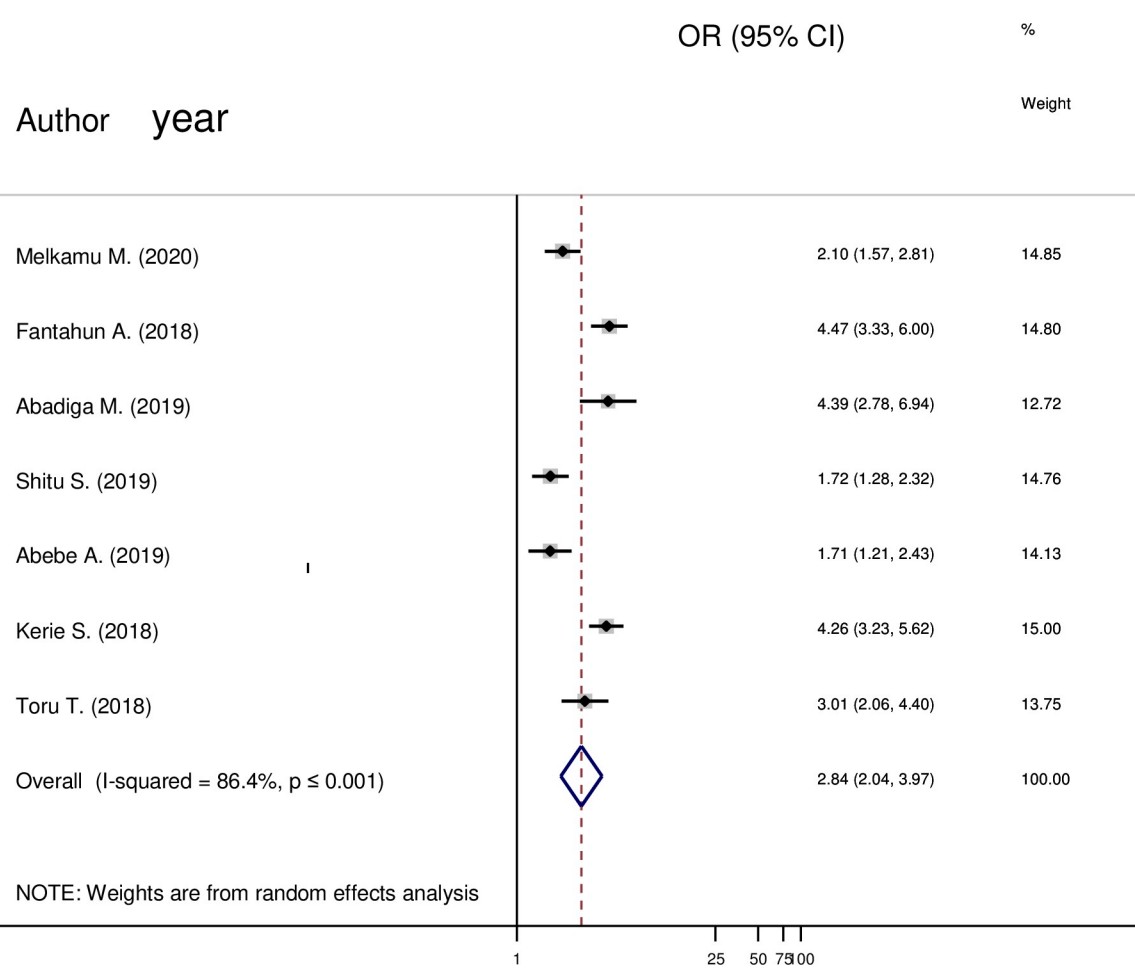

**Fig 6. Forest plot pooled random effect size (OR) of unplanned pregnancy-related to planned pregnancy in post-partum depression among mothers in Ethiopia, 2020.**

## Discussion

The pooled prevalence of post-partum depression in Ethiopia in our meta-analysis was 22.08%, with a 95% CI (17.66%, 26.49). Post-partum depression is strongly linked to life stress events (socio-economic factors), physical and emotional demands of childbearing and caring for new babies, and change in hormones after delivery [49]. In low and middle-income countries, post-partum depression is even more prevalent; the reason might be biological factors like (illness, and biological predisposition); psychosocial aspects (greater exposer to violence, the difficulty of living conditions, childhood maltreatment, social exclusion, and unplanned pregnancy), and economical (poverty and food insecurity) [10]. Another study showed that individuals who are living in low-income countries experienced more stressors associated with depression and anxiety than high-income countries [50].

This finding was consistent with other systematic findings. A systematic review of studies in 56 countries showed that the pooled prevalence of PPD was 17.7% [15]. In India, the systematic review and metanalysis, the pooled prevalence of PPD was 22% [16]; in another review, the pooled prevalence was 20% [17]. In Iran, a systematic review and meta-analysis, the pooled prevalence was 25.3 [18]. These estimates in low income and middle-income countries are similar to this finding, and taken together; they support a disagreement for

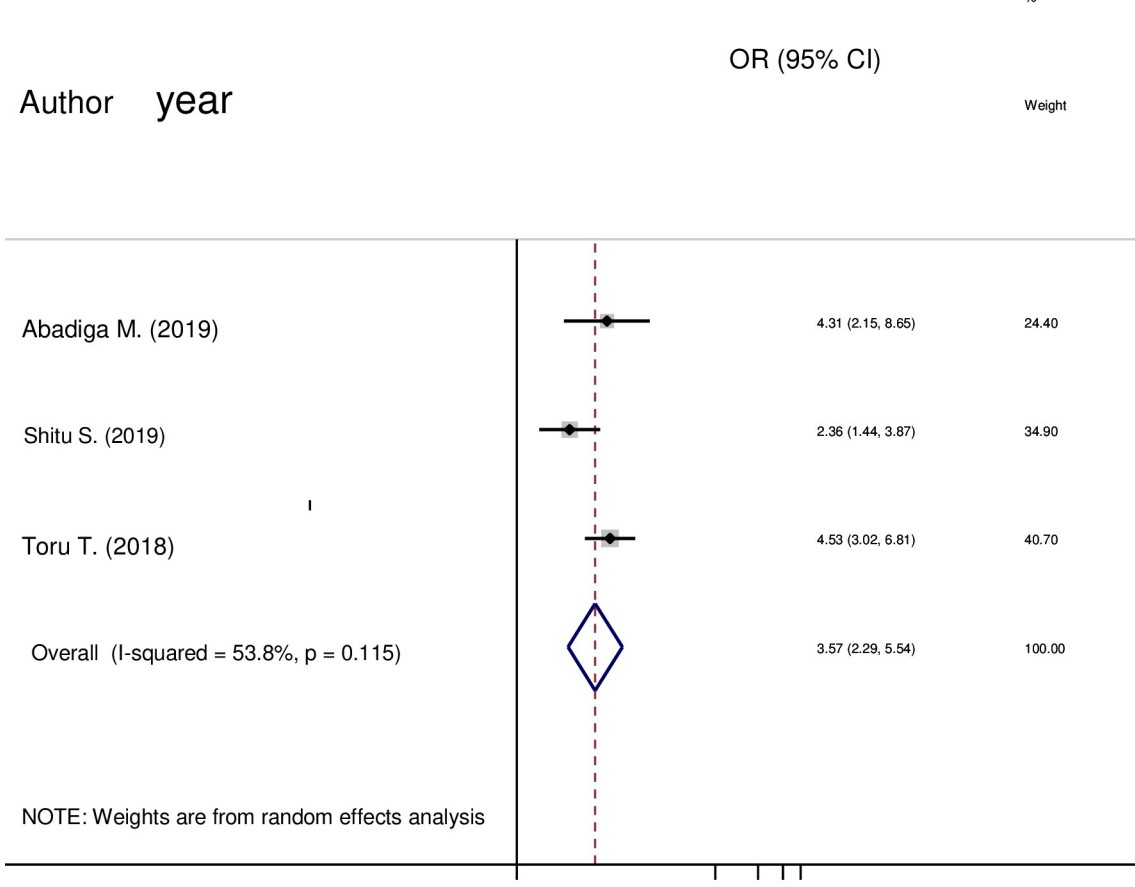

**Fig 7. Forest plot pooled random effect size (OR) of poor social support related to strong social support in post-partum depression among mothers in Ethiopia, 2020.**

placing more importance on maternal post-partum depression as part of overall efforts to maternal, infant, and child health. However, the current finding was significantly higher than the prevalence of 15% derived from a systematic review of studies from Denmark [19]. The discrepancy might be due to the absence of an awareness of PPD by health experts; there are issues that may be barriers to early recognition and management of post-partum depression [51–53].

Article review revealed that the following factors affect post-partum depression: unplanned pregnancy, social support, and domestic violence.

Postnatal mothers who had poor social support were about 4 [OR = 3.57; 95% CI (2.29,5.54)] times more likely to have depression when compared to mothers who had strong social support. Poor social support refers to perceived lack of support from husband [25], poor social support from the partner [26], poor social support [9,23,32], lack of support from the family [16,22] and dissatisfaction from family [28]. The reason might be that living with a supportive person halves stress. Social support is significant for maintaining good mental and physical health, and it related to resilience [54]. Good social support has been shown to be a consistent protective factor for mothers with high distress. The mothers who reported consistent opportunities to interact and talk with people were more likely to report a reduction in distress [55].

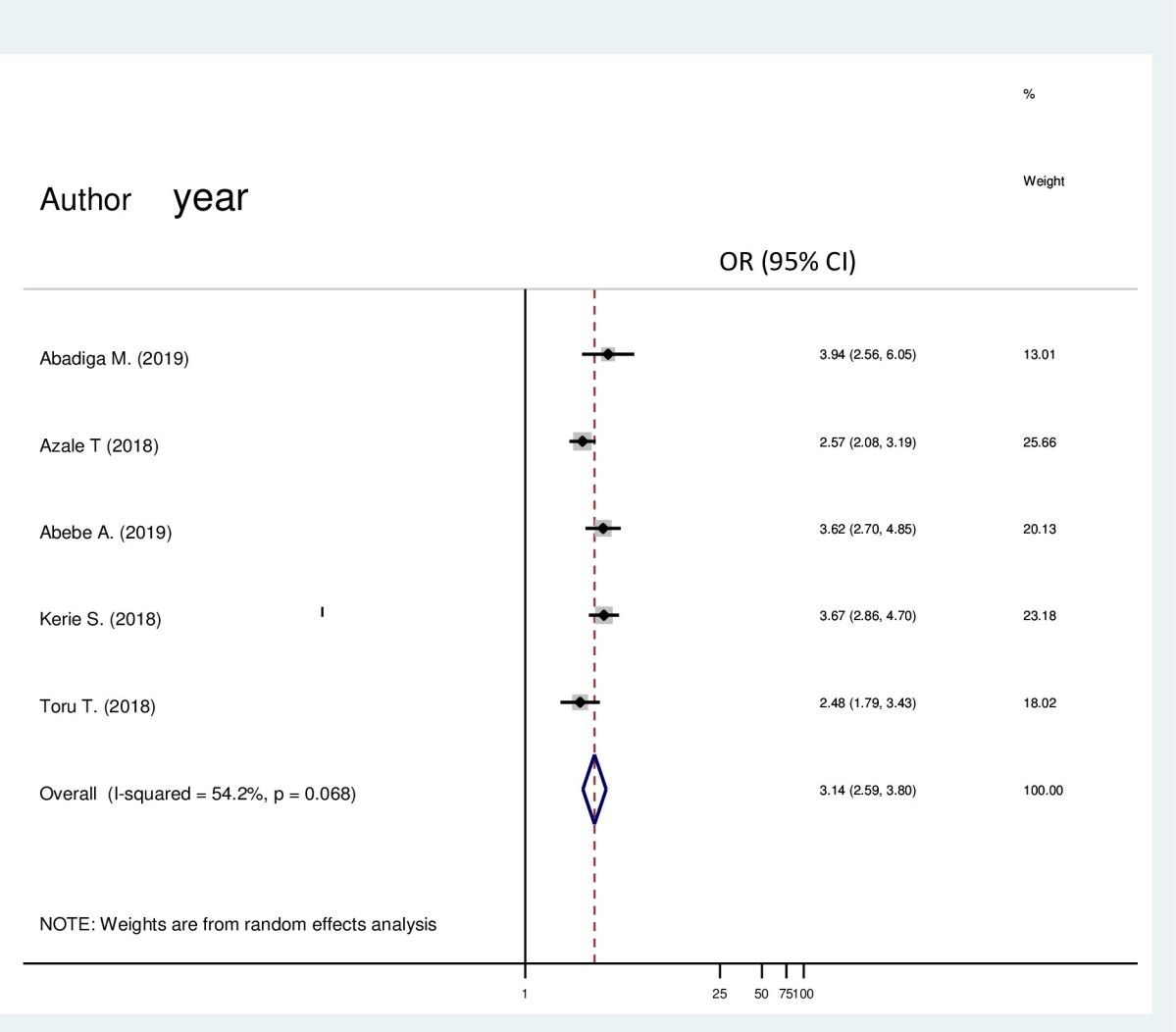

**Fig 8. Forest plot pooled random effect size (OR) of domestic violence related to no domestic violence in post-partum depression among mothers in Ethiopia, 2020.**

Mothers who had a history of domestic violence were about 3 [OR = 3.14; 95% CI (2.59,3.80)] times more likely to have depression than their counterparts. Other studies revealed that mothers who had a history of violence and abuse from husband [29–31] and domestic violence from the family members [16,22] were associated with post-partum depression. Violence has a negative effect on mental health. Mothers who are violent had feelings of helplessness and then had depression; they are also high risk for suicide to end their life [56].

Women whose pregnancy was unplanned were 3[(OR = 2.84; 95% CI (2.04, 3.97)] times more likely to have depression when compared with mothers who gave birth after a planned pregnancy. Having unplanned pregnancy, unwanted delivery, and unwanted pregnancy were factors affecting post-partum depression [20–22]. The reason might be during the conception period; psychological preparedness is very important to the mother, otherwise there will be mental health problems. Unplanned pregnancy leads the mother to feel unhappy and have

negative thoughts, consequently depression develops [57]. Mothers e with unplanned pregnancy had the earliest parenting stress [58]. The mother also perceived poor social support and less satisfaction with marriage life [59].

## Strengths and limitations

In this study, the authors used different databases to search the articles to minimize reviewers' bias and quality evaluation by four reviewers. Showing estimated pooled prevalence and pooled associated factors and conducting subgroup analysis based on assessment tools.

**Limitation.** In this study, only English language articles were included. A disproportional number of studies were included in the subgroup analysis of screening tools that minimize the estimated value's precision.

## Implication of this finding

The finding has implications for the future researcher, clinicians, and policymakers; for the future researcher, the prevalence of post-partum depression is increasing in the current finding. Therefore, it needs further investigation to know the reason why PPD is increasing and better management. Clinicians should screen the mother for depression when they present in the institution and the community. Our study should assist policymakersin design prevention and treatment strategies both in the community and in health institution.

**Conclusion.** Postpartum depression was high in Ethiopia. Unplanned pregnancy, poor social support, and domestic violence were factors affecting PPD. Therefore, the Ethiopian policymakers and health personnel better emphasize mothers who had a history of unplanned pregnancy, domestic violence, and poor social support.

## Supporting information

**S1 Checklist.**
(DOC)

**S1 File.**
(DOCX)

## Acknowledgments

The authors would like to thank all authors of the research paper included in this systematic review and meta-analysis.

## Author Contributions

**Conceptualization:** Tadele Amare Zeleke.

**Data curation:** Wondale Getinet, Zemenu Tadesse Tessema.

**Formal analysis:** Wondale Getinet, Zemenu Tadesse Tessema.

**Investigation:** Tadele Amare Zeleke, Wondale Getinet.

**Methodology:** Tadele Amare Zeleke, Wondale Getinet.

**Resources:** Zemenu Tadesse Tessema.

**Software:** Zemenu Tadesse Tessema.

**Supervision:** Kassahun Gebeyehu.

**Validation:** Kassahun Gebeyehu.

**Writing – original draft:** Tadele Amare Zeleke.

**Writing – review & editing:** Tadele Amare Zeleke, Kassahun Gebeyehu.

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
