## [Decision Letter · Decision Letter 0]

27 Oct 2020

PONE-D-20-11869

Prevalence and associated factors of postpartum depression in Ethiopia. A systematic review and meta-analysis, 2020

PLOS ONE

Dear Dr. Zeleke,

Thank you for submitting your manuscript to PLOS ONE. After careful consideration, we feel that it has merit but does not fully meet PLOS ONE’s publication criteria as it currently stands. Therefore, we invite you to submit a revised version of the manuscript that addresses the points raised during the review process.

The manuscript has been evaluated by two reviewers, and their comments are available below.

The reviewers have raised a number of concerns that need attention. They request additional information on methodological aspects of the study (such as the inclusion of information on the sample size and response rate), revisions to the statistical analyses and reporting, as well as revisions for English usage.

Could you please revise the manuscript to carefully address the concerns raised?

We look forward to receiving your revised manuscript.

Kind regards,

Vanessa Carels

Staff Editor

PLOS ONE

Journal Requirements:

4. Thank you for stating the following financial disclosure: 'no'

*Please include your amended statements within your cover letter; we will change the online submission form on your behalf.*

5. Thank you for stating the following in your Competing Interests section: 'no'

a. Please complete your Competing Interests statement to state any Competing Interests. If you have no competing interests, please state "The authors have declared that no competing interests exist.", as detailed online in our guide for authors at http://journals.plos.org/plosone/s/submit-now

6. Please include captions for your Supporting Information files at the end of your manuscript, and update any in-text citations to match accordingly. Please see our Supporting Information guidelines for more information: http://journals.plos.org/plosone/s/supporting-information

Reviewers' comments:

Reviewer's Responses to Questions

**Comments to the Author**

1. Is the manuscript technically sound, and do the data support the conclusions?

Reviewer #1: Yes

Reviewer #2: Yes

2. Has the statistical analysis been performed appropriately and rigorously? 

Reviewer #1: Yes

Reviewer #2: I Don't Know

3. Have the authors made all data underlying the findings in their manuscript fully available?

Reviewer #1: Yes

Reviewer #2: Yes

4. Is the manuscript presented in an intelligible fashion and written in standard English?

Reviewer #1: Yes

Reviewer #2: Yes

5. Review Comments to the Author

Reviewer #1: Title: Prevalence and associated factors of postpartum depression in Ethiopia. A systematic review and meta-analysis, 2020

Dear editor,

Thank you for giving me the opportunity to review this paper.

In this study, the authors conducted a systematic review and meta-analysis of the Prevalence and associated factors of postpartum depression in Ethiopia. This is an important area of research. The paper will be a useful addition to the literature and hopefully further research. Meanwhile, there are a lot of minor issues that need to be addressed for publication. I have briefly summarized them below

1. The review lacks detaildnesss in most of the contents of the reach, most importantly the results and discussion. Please elaborate on the results, discussion, methods, and other components.

2. Could you please remove 2020 from the title

3. Could you please revise the background section of the abstract in more convincing and succinct ways?

4. In the result section of the abstract, please indicated the analysis method used and highlight whether the factors are based on the meta-analysis results.

5. In the conclusion section of the abstract, just focus on the findings of the current study and it is advised to remove the sentence regarding the general population prevalence since you did not measure it.

6. In the introduction section, could you please revise the sentence regarding the definition of PPD, is that serious mental illness?

7. In the first paragraph of the introduction, could you please indicate the global epidemiology first and then go to PPD. Also, avoid repeated use of the word globe.

8. Generally, the introduction needs to be more capitalized and the justification section also needs to be included.

9. Please elaborate on the method section.

10. Please move the sentence regarding the results of publication bias from the method section to the result sections.

11. In the result section, please elaborate on the search process, and also include the detailed description of the characterizes of the included studies.

12. In the results of meta-analysis, specify whether a fixed or random-effect meta-analysis was used.

13. In table 1, please revise the author's name accordingly.

14. In the results section could you please included results of publication bias and sensitivity analysis (if heterogeneity exists).

15. Please more elaborate on the discussion. Also, include reasons for higher prevalence rates of depression in postpartum periods.

16. Please indicate the implication of the findings in the discussion section.

17. Elaborate on the strength and limitations of the study.

18. Revised conclusion section. Also, remove sentence regarding the general population prevalence, focus on your findings.

Reviewer #2: 1. Abstract:

Line 2: most common complication

2. Introduction

Para 1: revise language

3. Para 3: separate low and middle income countries from high income countries. I respectfully suggest

4. Page 5, para 2: globally? LMIC? HIC? Clarify please

Methods

6. Independent variables: You mention as factors associated with postpartum depression “D were unplanned pregnancy(16-18), a having history of depression(10, 19-21) , perceived lack of support from husband(21), domestic violence and lack of support(8, 18), poor social support from the partner(22), birth complication(23), dissatisfaction about family(24), violence from husband(25-27) and poor social support(4, 19, 28).”. However you only examine poor v strong social support, pregnancy intendedness and exposure to violence. These turn out to significant associations. I suggest you test and report all the previously mentioned factors, and report the negative results (if that should be the case)

Results:

7. Features of the studies, “comprised” maybe “carried out”?

Discussion:

8. The reason there is a difference between pooled postpartum depression in LMIC and HIC may also be related to factors such as social exclusion, poverty, food insecurity, greater exposure to violence etc. I suggest you look at Herba et al (Lancet Psychiatry 2016 Oct;3(10):983-992.) doi: 10.1016/S2215-0366(16)30148-1. Epub 2016 Sep 17 and other recent literature to iMprove this part of the discussion.

9. Please elaborate on the effect of social support on maternal mental health. “The reason might be due to individuals who has stress, share their stress to another person, stress reduced by half.” Is not enough

10. Please elaborate on the untoward impact of violence on mental health (via chronic stress, for instance) there is a lot of literature on this topic.

11. please elaborate on the potential mechanism of unwanted pregnancy as a risk factor for postpartum depression.

11. English needs some revision by a native or near native speaker.

6. PLOS authors have the option to publish the peer review history of their article (what does this mean?). If published, this will include your full peer review and any attached files.

Reviewer #1: **Yes: **Getinet Ayano

Reviewer #2: No

---

## [Author Response · Author response to Decision Letter 0]

24 Nov 2020

Response to Reviewers’

Dear Reviewer1

Thank you for your valuable comments and questions. Authors tried to amend the comments and answering questions accordingly. The detailed is in the “manuscript track change”

Sincerely,

Tadele Amare

1. The review lacks detailedness in most of the contents of the reach, most importantly the results and discussion. Please elaborate on the results, discussion, methods, and other components.

Response: it is elaborated in each section accordingly. 

2. Could you please remove 2020 from the title

Response: it is removed page1 line 2

3. Could you please revise the background section of the abstract in more convincing and succinct ways?

Response: the background is revised as convincing page 24 to 27

4. In the result section of the abstract, please indicated the analysis method used and highlight whether the factors are based on the meta-analysis results.

Response: random effect size model was used and factors were analyzed in meta-analysis. Page2 line 43 to 44

5. In the conclusion section of the abstract, just focus on the findings of the current study and it is advised to remove the sentence regarding the general population prevalence since you did not measure it. 

Response: it is removed. Page3 line 48 to 49

6. In the introduction section, could you please revise the sentence regarding the definition of PPD, is that serious mental illness?

Response: it is modified as “Postpartum depression (PPD) is a mental illness that involves the brain and affects behavior and physical health after delivery” Page 5 line 103 to 104

7. In the first paragraph of the introduction, could you please indicate the global epidemiology first and then go to PPD. Also, avoid repeated use of the word globe.

Response: it is modified in page 5 line 96 to102

8. Generally, the introduction needs to be more capitalized and the justification section also needs to be included.

Response: the introduction section is modified and justification section is also added page 4 to 7 and line 65 to 138

9. Please elaborate on the method section.

Response: it is elaborated page 8 to 11 line 151 to 228

10. Please move the sentence regarding the results of publication bias from the method section to the result sections.

Response: it is moved from method section to result section page 15 line 274 to 276

11. In the result section, please elaborate on the search process, and also include the detailed description of the characterizes of the included studies.

Response: it is elaborated and characterized page 12 to 14 line 239 to 273

12. In the results of meta-analysis, specify whether a fixed or random-effect meta-analysis was used.

Response: the random effect model was used and corrected in page 15 line 291 to 292

13. In table 1, please revise the author's name accordingly.

Response: Tabel1 and Table2 the author’s name is corrected Table1 page 13, line 261 to 262 and Table2 page 14 line 272 to 273. 

14. In the results section could you please included results of publication bias and sensitivity analysis (if heterogeneity exists).

Response: publication bias and sensitivity analysis is included. Publication bias page 15 line 274 to 276 (see also Fig2 in figure section) and sensitivity test page15 line 279 to 281

15. Please more elaborate on the discussion. Also, include reasons for higher prevalence rates of depression in postpartum periods.

Response: the discussion section is modified page 18 to 21 line 340 to 410 

16. Please indicate the implication of the findings in the discussion section.

Response: implication of the finding is stated in page 21 line 420 to 426

17. Elaborate on the strength and limitations of the study.

Response: it is elaborated in page 21 line 412 to 418

18. Revised conclusion section. Also, remove sentence regarding the general population prevalence, focus on your findings. 

Response: it is revised and removed “general population”. Page 21 to 22 line 427 to 431

Dear Reviewer2,

Thank you for your valuable comments and questions. Authors tried to amend the comments and answering questions accordingly. The detailed is in the “ manuscript track change”

Sincerely,

Tadele Amare

Reviewer #2: 1. Abstract:

Line 2: most common complication

Response: it is modified page 2 line 24 to 27

2. Introduction

Para 1: revise language

Response: it is totally modified page 5 line 103 to 104 

3. Para 3: separate low- and middle-income countries from high income countries. I respectfully suggest

Response: it is separated in page5 line 96 to 102 and again in page 6 line 118 to 126

4. Page 5, para 2: globally? LMIC? HIC? Clarify please

Response: para2. This is the global study page6 line 108. 

Methods

6. Independent variables: You mention as factors associated with postpartum depression“ were unplanned pregnancy(16-18), a having history of depression(10, 19-21) , perceived lack of support from husband(21), domestic violence and lack of support(8, 18), poor social support from the partner(22), birth complication(23), dissatisfaction about family(24), violence from husband(25-27) and poor social support(4, 19, 28).”. However, you only examine poor vs strong social support, pregnancy intendedness and exposure to violence. These turn out to significant associations. I suggest you test and report all the previously mentioned factors, and report the negative results (if that should be the case)

Response: These all factors were found in the literature review in different countries other than Ethiopia. The factors that were included in meta-analysis were conducted in Ethiopia. 

Results:

7. Features of the studies, “comprised” maybe “carried out”?

Response: it is rewrite page12 line 251 to 256

Discussion:

8. The reason there is a difference between pooled postpartum depression in LMIC and HIC may also be related to factors such as social exclusion, poverty, food insecurity, greater exposure to violence etc. I suggest you look at Herba et al (Lancet Psychiatry 2016 Oct;3(10):983-992.). Epub 2016 Sep 17 and other recent literature to iMprove this part of the discussion.

Response: it is added in page 18 line 342 to 351

9. Please elaborate on the effect of social support on maternal mental health. “The reason might be due to individuals who has stress, share their stress to another person, stress reduced by half.” Is not enough

Response: other justifications are added page 19 and 20 line 383 to 387

10. Please elaborate on the untoward impact of violence on mental health (via chronic stress, for instance) there is a lot of literature on this topic.

Response: it is elaborated page 20 line 388 to 398

11. please elaborate on the potential mechanism of unwanted pregnancy as a risk factor for postpartum depression.

Response: it is elaborated in page 20 and 21 line 399 to 410

11. English needs some revision by a native or near native speaker.

Response: the English language is edited by the English language experts at Department of English language and literature, College of Social Science and Humanities, University of Gondar.

---

## [Editor Report · Decision Letter 1]

25 Jan 2021

PONE-D-20-11869R1

Prevalence and associated factors of postpartum depression in Ethiopia. A systematic review and meta-analysis

PLOS ONE

Dear Dr. Zeleke,

Thank you for submitting your manuscript to PLOS ONE. After careful consideration, we feel that it has merit but does not fully meet PLOS ONE’s publication criteria as it currently stands. Therefore, we invite you to submit a revised version of the manuscript that addresses the points raised during the review process.

I see that you have introduced the changes and answered the queries, and I thank you for your dedicated review of the language. However it is still not ready, Please look at the attached Word version with some amends and accept them, so that we may go on with this process.

This paper will be a useful addition to the literature on perinatal depression, particularly for practitioners and policymakers in  LMIC. Please make a last effort. 

We look forward to receiving your revised manuscript.

Kind regards,

Marta B Rondon, M.D.

Academic Editor

PLOS ONE

Additional Editor Comments (if provided):

thank you for your dedicated review of English. However, it is not ready yet,

Please look at the amended Word files I am enclosing.

---

## [Author Response · Author response to Decision Letter 1]

26 Jan 2021

Response for reviewers’

Thank you for your commitment in reviewing of the manuscript. Based on the given comments, the English language is edited point by point.

Sincerely, 

Tadele Amare

---

## [Decision Letter · Decision Letter 2]

1 Feb 2021

Prevalence and associated factors of postpartum depression in Ethiopia. A systematic review and meta-analysis

PONE-D-20-11869R2

Dear Dr. Zeleke,

We’re pleased to inform you that your manuscript has been judged scientifically suitable for publication and will be formally accepted for publication once it meets all outstanding technical requirements.

Kind regards,

Marta B Rondon, M.D.

Guest Editor

PLOS ONE

Additional Editor Comments (optional):

You have answered our queries and have satisfied our observation. Pleased to see this accepted and even more pleased when I see this in print. Kudos.

Reviewers' comments:

Reviewer's Responses to Questions

**Comments to the Author**

1. If the authors have adequately addressed your comments raised in a previous round of review and you feel that this manuscript is now acceptable for publication, you may indicate that here to bypass the “Comments to the Author” section, enter your conflict of interest statement in the “Confidential to Editor” section, and submit your "Accept" recommendation.

Reviewer #3: All comments have been addressed

2. Is the manuscript technically sound, and do the data support the conclusions?

Reviewer #3: Yes

3. Has the statistical analysis been performed appropriately and rigorously? 

Reviewer #3: Yes

4. Have the authors made all data underlying the findings in their manuscript fully available?

Reviewer #3: Yes

5. Is the manuscript presented in an intelligible fashion and written in standard English?

Reviewer #3: Yes

6. Review Comments to the Author

Reviewer #3: The authors have adequately addressed my previous concerns. The findings of the study will contribute to the limited evidence in low and middle-income countries.

7. PLOS authors have the option to publish the peer review history of their article (what does this mean?). If published, this will include your full peer review and any attached files.

Reviewer #3: No

---

## [Editor Report · Acceptance letter]

3 Feb 2021

PONE-D-20-11869R2 

Prevalence and associated factors of post-partum depression in Ethiopia. A systematic review and meta-analysis. 

Dear Dr. Zeleke:

I'm pleased to inform you that your manuscript has been deemed suitable for publication in PLOS ONE. Congratulations! Your manuscript is now with our production department. 

Kind regards, 

on behalf of

Dr. Marta B Rondon 

Guest Editor

PLOS ONE